# A global open-source database of flood-protection levees on river deltas (openDELvE)

Jaap H. Nienhuis[1], Jana R. Cox[1], Joey O'Dell[1], Douglas A. Edmonds[2], and Paolo Scussolini[3]

[1] Department of Physical Geography, Universiteit Utrecht, Postbus 80.115, 3508 TC Utrecht, Netherlands
[2] Department of Earth and Atmospheric Sciences, Indiana University Bloomington, 1001 East 10th Street, Bloomington, IN 47405-1405, United States of America
[3] Institute for Environmental Studies, Vrije Universiteit Amsterdam, De Boelelaan 1111, 1081 HV Amsterdam, Netherlands

*Correspondence to*: Jaap H. Nienhuis (j.h.nienhuis@uu.nl)

**Abstract.** Flood-protection levees have been built along rivers and coastlines globally. Current datasets, however, are generally confined to territorial boundaries (national datasets) and are not always easily accessible, posing limitations for hydrologic models and assessments of flood hazard. Here, we bridge this knowledge gap by collecting and standardising global flood-protection levee data for river deltas into the **open**-source global river **de**lta **lev**ee data **e**nvironment, openDELvE. In openDELvE, we aggregate levee data from national databases, reports, maps, and satellite imagery. The database identifies the river delta land areas that the levees have been designed to protect. Where data are available, we record the extent and design specifications of the levees themselves (e.g., levee height, crest width, construction material) in a harmonised format. The 1,657 polygons of openDELvE contain 19,248 km of levees and 44,733.505 km$^2$ of leveed area. For the 153 deltas included in openDELvE, 17% of their land area is confined by flood-protection levees. Around 26% of delta population lives within the 17% of delta area that is protected, making leveed areas densely populated. openDELvE data can help improve flood exposure assessments, many of which currently do not account for flood-protection levees. We find that current flood hazard assessments that do not include levees may exaggerate the delta flood exposure by 33% on average, but up to 100% for some deltas. openDELvE is made public on an interactive platform (www.opendelve.eu), which includes a community-driven revision tool to encourage inclusion of new levee data and continuous improvement and refinement of open-source levee data.

## 1 Introduction

### 1.1 What are levees and what do they do?

Levees are banks of sediment or artificial material that prevent water from entering areas where it is not desirable. They are common in delta plains and protect their populations and assets from water level fluctuations of rivers and the sea. Levees have been constructed to mitigate flood risk and direct water flows throughout human civilisation. Recorded building of levees along the River Nile in Egypt began around 4600 BP (Westermann, 1919) which indicates the innate link between the settlement of coastal populations and the development of levees. Modern materials and engineering concepts have altered the overall appearance and effectiveness of levees, but the basic principle has remained the same for millennia.

Levees can also have negative environmental consequences. They alter sediment transport and sedimentation patterns, as sediment deposition behind levees is usually reduced. Areas protected by levees can subside relative to the surrounding (Middelkoop et al., 2010), resulting in increased risk of coastal and river flooding in the longer term (Pinter et al., 2008; Criss and Shock, 2001; Pinter, 2005; Munoz et al., 2018). In particular, leveed deltas are at risk to be locked-in (Santos and Dekker, 2020), as areas become sediment-starved and cease to keep up with sea level rise (Pinter et al., 2016). Another example of the

negative effect of levees is in Australia, where undocumented private levees, intending to protect land, resulted in degradation of the floodplain ecosystem, and contributed to flash flood risk by disconnecting floodplain and channel (Steinfeld et al., 2013).

Because of the negative consequences, contemporary river and flood management measures/projects often prioritise nature-based solutions that limit the need for levees (Esteves, 2014; Cohen-Shacham et al., 2016; Van Wesenbeeck et al., 2014). In deltas in particular, levees are sometimes removed to pursue sedimentation-enhancing strategies (Cox et al., 2022) that restore natural delta functions, but this may not always be possible.

**1.2 Why (data of) levees matter**

Data on levees are important, especially for river deltas. People living in river deltas face mounting threats: they are disproportionately affected by coastal flooding and relative sea level rise (Edmonds et al., 2020) and rely on river sediment supply that is diminishing in many places (Dunn et al., 2019). Data on levees can help to assess these threats.

Mapping the presence of levees is useful for hydrologic and hydrodynamic modelling. Such models are used to predict
inundation during high water levels in rivers or in the sea, and help active management of risk and hazard to life. Models are also used to design levees by simulating a specific flood return period or flood scenario. Levees can be incorporated in detailed models (e.g., HEC-RAS, US Army Corps of Engineers, 2020; or Delft3D, Lesser et al., 2004) as a geometric feature within an initial surface topography. For models on larger scales, levees are too small to be included directly and are sometimes presented as a sub-grid feature or through a flood-attenuation proxy (Sampson et al., 2015).


Data on levees can help to better understand human-landscape interactions (Werner and McNamara, 2007). One of these interactions is the so-called "levee effect", defined by Gilbert White in 1947, whereby levee building creates an excessive sense of security which leads to increased development and increased flood exposure (Hutton et al., 2019). This effect of levees is thought to contribute to larger exposure to low-probability floods in delta cities. New Orleans after Hurricane Katrina is an
example (Kates et al., 2006). Data on levees can help to assess the co-evolution of levees and development prior to, and in response to disasters, both present and future, and better understand the levee effect (Di Baldassarre et al., 2018).

Levee data can also help studies on levee failures, which are a globally significant source of flood risk. Özer et al. (2019) have developed the International Levee Performance Database (ILPD), presenting data on levee testing and failure events in an
interactive and searchable interface. Levee data for hazard assessment purposes is additionally useful outside the realm of geophysical modelling, and is core to civil engineering and emergency response management for levee performance, such as during the safety and risk calculation of hurricanes (Mitchell et al., 2013). Data can also be relevant for large-scale studies into the effects and costs of levees, and in their comparison with alternative flood-risk reduction strategies in these areas (Ibáñez et al., 2014; Scussolini et al., 2017; Vuik et al., 2019; Cox et al., 2022). The insurance industry, local residents, and homeowners
are additional users of levee data and modelling outputs that may help with their hazard and risk assessments (National Research Council, 2013).

**1.3 A (data) gap in levees**

Despite the potential use of levee data, locations and characteristics of levees are often poorly documented (Scussolini et al., 2016; Özer et al., 2019), resulting in inaccuracies and challenges for flood risk modelling (Sampson et al., 2015; Trigg et al.,
2016; Winsemius et al., 2016; Dullaart et al., 2021), hazard modelling (Di Baldassarre et al., 2009), and sea-level rise impact modelling (Nienhuis and van de Wal, 2021). Accurate models require data input about levees including their spatial extent, protected area, and basic attributes, which currently does not exist in a coherent and harmonised single geospatial data format.

Levees themselves are not new creations, and so most data that references their locations and standards is historical and recorded in paper form (maps, plans etc.). It is typically governments and municipal organisations who plan and construct levees. These institutions (e.g., USACE) also maintain them as part of their daily operations and produce maps and datasets about their design, operation, and failure. This gives a plethora of data such as reports and design specifications, which allows for accurate data gathering and collection processes without the need for in-person observation (e.g., USACE National Levee Database, levees.sec.usace.army.mil). Generally, this results in good quality central national databases, sometimes complemented by higher resolution regional variants (e.g., New South Wales' Distinctive Land Surface Dataset, Australia), but they rarely extend past administrative borders. Data availability can also be publicly restricted.

Poor data on levee existence and levee properties have made it such that their presence is often disregarded in global flood modelling (Trigg et al., 2016) and global delta modelling (Nienhuis et al., 2020). The lack of levee data results in suboptimal modelling results (Fleischmann et al., 2019). The WRI AQUEDUCT Global Flood Analyzer is an example (https://www.wri.org/data/aqueduct-global-flood-analyzer). It provides exceptional global-level flood hazard data but does not include levees and results in overpredicted flood exposure for heavily leveed areas such as the Netherlands.

While specific aspects of levee failure have been documented and aggregated globally (i.e., Özer et al., 2019), we are not aware of any open-source approaches that collect, harmonise and attribute information on levee extent. The lack of global registration of levees complicates flood management efforts. As an alternative, FLOPROS (Scussolini et al., 2016) presents a global dataset on existing and policy-level flood protection standards. This implicitly includes the flood protection offered by levees, but does not include data on levees. Other approaches exist that use (semi-)automated algorithms to locate and specify levees from LIDAR data (e.g. Steinfeld et al., 2013; Wing et al., 2019) but this is restricted by data availability and is not yet possible globally. A levee database can help inform those algorithms and provide validation and calibration data. Besides the registration of their existence of levees, communication and awareness of this information is important, to enable the above-listed uses of levee information.

### 1.4 Objective

The objective of openDELvE is to provide an source of delta levee protection data, for both primary use in flood and hazard modelling, as well as secondary community use through increased data availability by publishing the data on a public website (http://www.opendelve.eu) following standardized data types. openDELvE includes links to original data sources, as well as a user-led amendment reporting function. Examples are also given of openDELvE use for hazard modelling and delta modelling improvements.

## 2 Methods

### 2.1 Overview

openDELvE is a collection of existing data on levees and protection features on deltas. We have collected data from vector, raster, and documentary sources. This resulted in two geospatial layers – one for levees, and one for leveed areas – and a supporting index dataset, linked to the respective delta by a unique identifier and cross-mapped to the river delta dataset of Edmonds et al. (2020). Our methods allow for replicable tracing, processing, assimilation, and display of the data. By storing individual level references and assessing data quality, we aim to provide data that is open and transparent. Our work is underpinned by the principles of FAIR science to support reuse by producing data that is Findable, Accessible, Interoperable,

and Reusable (Wilkinson et al., 2016). openDELvE development followed these steps: data definition (Sect. 2.2), data collection (2.3), data processing (2.4), data attribution (2.5), data management (2.6), and data assurance (2.7).

**2.2 Data definition**

We followed our definition of levees from Sect. 1.1. Levees exist along coasts and rivers globally, but the scope of openDELvE is limited to river deltas (Sect. 2.4.1). We made use of a database of deltaic locations and deltaic area extent by Caldwell et al. (2019) and Edmonds et al. (2020). We further limited ourselves to only storing information on defences that are permanent features, and not temporary/reactive measures. Sandbags and hoardings deployed for flash flooding or imminent but irregular flood issues are temporary, and so are usually not mapped, nor were considered for inclusion in this database.

openDELvE is designed to represent levees as geospatially explicit vector data: lines and polygons. For source data that exists in reports on maps and technical drawings levee presence is often reduced to a raster map element, and so needed to be sufficiently georeferenced and assessed for quality. However, we still consider this a valid data source and have included it in our process. We consider the age, source document, and data quality as we recognise that data may be reworked and requoted
a number of times in its lifespan.

openDELvE consists of three data elements: an index table and two vector layers (Table 1), each with a set of standardised attributes (Table 2). Data itself include a data quality class and a direct link to the source dataset. We devised the data quality criteria included in Table 3:


**Table 1: Data entities in the live viewing environment and their exported file types as in the research data store**

| Data Entity | Type | Exported Elements | Purpose |
|---|---|---|---|
| Delta Index | Table | CSV | Contains data decision logs and linking levees at delta level |
| Leveed Area | Polygon | SHP, KML | A vector layer containing polygons of the areas protected by levees |
| Levee Lines | Line | SHP, KML | A vector layer containing lines of the levees and including standardised attributes |

**Table 2: openDELvE attributes for the three data elements (as per Table 1). Conversion factors and mapping of fields are given in Supplementary Table S2.**

| Data Entity | Attribute | Purpose |
|---|---|---|
| **Delta Index** | FriendlyName | Name of the delta, if known |
| | Status | Processed, No Result, Pending, or Not Processed (as per Sect. 2.3) |
| | PolygonID | Delta ID following Edmonds et al. (2020) |
| | ISO_2 | 2-digit code identifying the country where the majority of the delta lies, following ISO 3166-1:2020 alpha-2 |
| | Journal | A timestamped text log of activity at a delta level |
| | MainRefAPA7 | Literature reference for the overall source material for the delta, formatted in APA 7th Edition, if available |
| | MainRefDOI | Digital Object Identifier for the source material, if available |
| | NeedsReview | Boolean indicator of requirement for later review of delta |
| | LastChkDate | Date field signalling last check date of the delta |
| | LastChkBy | Two-character identifier of the last user who updated the dataset |
| **Leveed Area** | NAME | Name of the leveed area feature from the source dataset, if available |

| | REFERENCE | The identifier for the feature from the source dataset, if available |
| --- | --- | --- |
| | DOI | Digital Object Identifier for the source material, if available |
| | URL | Uniform Resource Locator (web link) for the source material, if available |
| | LITREF | Literature reference for the source material, formatted in APA 7th Edition |
| | PolygonID | Delta ID following Edmonds et al. (2020) |
| | DataQuality | Data quality classification (following Table 3) |
| **Levee Lines** | NAME | The name for the feature from the source dataset, if available |
| | REFERENCE | The identifier for the feature from the source dataset, if available |
| | DOI | Digital Object Identifier for the source material, if available |
| | URL | Uniform Resource Locator (web link) for the source material, if available |
| | LITREF | Literature reference for the source material, formatted in APA 7th Edition |
| | DefenceLength | The length of the levee feature, as provided in the source dataset, if available (metres) |
| | DefenceHeight | The height of the levee feature, as provided in the source dataset, if available (metres) |
| | DefenceWidth | The width of the levee feature, as provided in the source dataset, if available (metres) |
| | FoundationWidth | The width of the levee foundation, as provided in the source dataset, if available (metres) |
| | Construction | The primary material that the levee is composed of |
| | ClassType | Construction or formation type of the feature |
| | CutoffMaterial | The material that the levee cutoff is composed of |
| | DesignStandard | Design storm rating of the feature (1/n, decimal) |
| | DataQuality | Data quality classification (following Table 3) |
| | PolygonID | Delta ID following Edmonds et al. (2020) |


**Table 3: Data quality definition for levees based upon data provenance, both for use in initial data classification and ongoing maintenance. Criteria are exclusively applied: all categories must be met to meet a certain classification.**

| Class | Criteria |
| --- | --- |
| A (Excellent) | Vector data |
| | First-order data source (i.e., scientific papers, governmental geospatial data, original publication) |
| | Spatially complete[a] (with respect to geopolitical boundaries) |
| | Existence verifiable with satellite imagery |
| B (Good) | Raster data (suitably georeferenced, little to no variance) |
| | First-order or re-cited/modified (original accessible) but published within a scientific or government publication |
| | Existence verifiable with satellite imagery |
| C (Acceptable) | Raster data (loosely georeferenced, variance due to old base map or similar) |
| | Conjectural or non-scientific source (ex: newspaper) |
| | Source >20 years of age, regardless of type |
| | Existence (partially[b]) verifiable with satellite imagery |
| X[c] (Invalid) | Data inaccessible (blocked, hidden, unpublished) |
| | Irrecoverable issues with data quality |
| | Could not confirm existence of data from other sources using satellite imagery with resolution $\leq$25m |
| | Temporary or reactive measures only (ex: sandbags) |

[a]*Data that were attributable to class X have not been included in the published dataset but are documented in the delta index*

[b]*We included 'partially verifiable' due to incident patchy local coverage of openly accessible satellite data, as there are instances where sufficient high-resolution imagery was not accessible, but standard-resolution imagery indicated the presence of the feature that was elsewhere published.*

$^c$*Spatially complete was defined as being of the entire levee run, which may be comprised of several subsection maps.*

### 2.3 Data collection

We conducted extensive literature searches using a variety of web searching platforms (i.e., Clarivate Web of Science, Google Search, Google Scholar, OCLC WorldCat) as well as data aggregation platforms (e.g., re3data.org, DataCite, data.gov.uk, data.gov, data.gov.au). Data was collected in a search process that is documented as a log with diary-style entries in the Delta Index table (see Table 1) and recorded at a delta level. Sources for each individual levee are stored at the feature level. This allowed us to record rationale and decision-making process so that both viewers and onward developers of the dataset are

aware of the steps taken and explanations for decisions taken in data hand.

With an international scope, searching often required country or location-specific terms (e.g., '*tanggul*' meaning levee or embankment in Indonesian) to aid data discovery, and these were regionally supplemented along with a vocabulary of common delta and levee terms when using academic paper and internet indexing services.


Funding reports from the World Bank projects on flood defence activities have also contributed to the database. Financing documents often contain maps and so we include data from the World Bank where it was discovered in our searches, released publicly, had been reviewed, and contained levee feature level data.

When it was not possible to find data in areas where levees were expected, the place was identified by name using the address search (*gazetteer*) function in ArcGIS and then basic internet searching was performed to find reports of floods or sea-level rise related damage. Finally, we made use of the world satellite imagery layer within ArcGIS to review areas where levee source data was inaccessible, and assess by visual means whether it was likely levees were present. We verified areas that we believe may be uninhabited using this imagery and classified them accordingly, where satellite imagery confirmed no visible

levees, the delta was set to 'No Result'. If levees were visible but we could not verify them with alternative data sources, we set the delta to 'Pending' where external enquiries were taking place and the relevant note was entered in the ArcGIS journal (see Table 2). We identify deltas as 'Not Processed' if we have yet to manually review available sources, and no national vector dataset was discoverable for processing via our automated tool.

Many deltas in the delta dataset may be small and uninhabited (Edmonds et al., 2020), have inaccessible data, or have data that we were unable to convert into a format that we could add to the database. We collectively group these deltas as having 'No result' in terms of data collection. Note that this does not always mean there is no data. For example, data from the Database nazionale della AgriNature in TErra (DANTE, *formerly known as: ItaliaN LEvee Database [INLED]*) (Barbetta et al., 2015) was not suitable for processing because it only contains a levee start and end point coordinate. We classified these

deltas under 'No result' because it requires access to a detailed regional-level watercourses database and high-resolution DEM so that an interpretational algorithm could be trained to infer the levee course.

Where available, we included levee attributes (e.g., design storm, wall height, levee material, Table 2). This can inform modelling and therefore work as a stand-alone spatial tool for investigating river delta dynamics. Additionally, the data layers

can be used for verification of deductive models for the detection of levees by other means, including LIDAR and remotely sensed data as well as corroborating other data sources, such as OpenStreetMap. As we intend for the database to be globally comparable, we set up a cross matching list (Supp. Table S2) within the project documentation to ensure that the attributes of the levee lines layer were consistent between sources and languages. This was then used for both manual and automated input so that different units of measure, classifications of levee and construction type, and key engineering data were harmonious.

**2.4 Data processing**

**2.4.1 Vector data processing**

Where data were sourced in vector format, we defined a data processing algorithm in the ArcGIS® Model Builder (Supp. Fig. S1) to clip the imported data to the extent of river deltas from Edmonds et al. (2020) with a 100 km 'buffer zone'. This buffer zone is included to maximize OpenDELvE data usability, but it does not affect reported statistics on delta coverage: all reported data statistics in this paper are for levees strictly within delta boundaries (Figure 1), although these can include shallow marine portions of the delta front as well as upland area (Figure 2, Edmonds et al., 2020). The buffer zone is included to allow extended use of the dataset for upstream fluvial and sediment transport modelling and additionally, should the dataset of Edmonds et al. (2020) be updated, reduces the likelihood that levees are missed from the layer.

The ArcGIS® Model Builder automated import process is distributed with the dataset so that data can be repeatedly processed and added to the database both now and in the future. We supplemented this by the creation of conversion tables *(*Supp. Table S2*)* so that levee attributes, where available, are comparable at a global scale.

**2.4.2 Non-vector data processing**

We performed georeferencing of levee maps where the location was visible using a second, georeferenced, map and the map could be referenced in fewer than 5 reference points. This ensured that we were not extensively distorting the source map and therefore it was possible for us to trace in the features as accurately as possible. Where no georeferencing within 5 reference points was possible, or where the map had too few defining features to be georeferenced at all (e.g. map created with too few topographical features, substantial engineered or geological change resulted in difference between map and modern day situation) then the appropriate data quality class (X) was assigned. The data source was set aside and the process was documented in the log. Furthermore, where aerial photography was analysed, we defined a set protocol for the inference of leveed area (Supp. Fig. S3).

Data in the "Levee Lines" layer is currently limited to vector levee data sources and does not exist for raster data sources. Ongoing work includes manual review and development of (semi) automated processing steps to retrieve levee lines from raster sources.

**2.4.3 Extraction of leveed-areas from levee line information**

Several data sources were processed where only levee lines are available, and not levee-protected areas (polygons). In these cases, we estimated levee-protected areas from levees by: (1) manually selecting levees that are not separated by water bodies, and (2) constructing an area confined by these levees (Figure 1). We manually reviewed this process using datasets that have both levee lines and leveed areas (e.g., USACE National Levee Database) and did not result in a large over- or under estimation of the leveed area (Fig. S3). Leveed area generated from this process instead of original data is indicated in the data quality label.

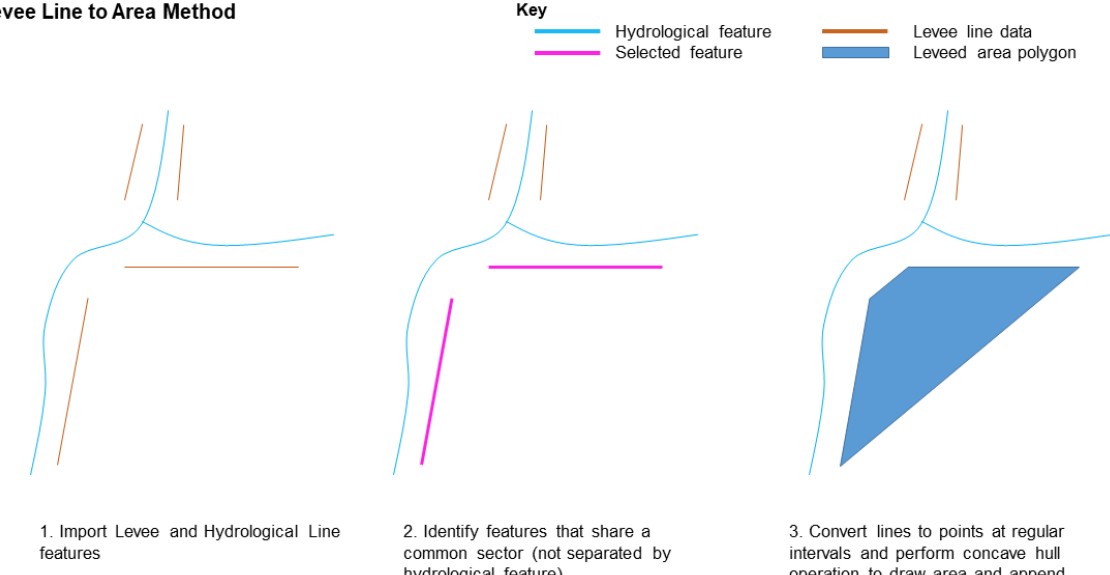

**Levee Line to Area Method**

**Key**
— Hydrological feature
— Selected feature
— Levee line data
▰ Leveed area polygon

1. Import Levee and Hydrological Line features

2. Identify features that share a common sector (not separated by hydrological feature)

3. Convert lines to points at regular intervals and perform concave hull operation to draw area and append

**Figure 1: Extraction of leveed-areas from levee line information (visual representation of process outline in Supplementary Figure S3)**

**2.5 Data attribution**

Every task performed was recorded in the openDELvE metadata (Delta Index, field: Journal (Table 2)  ) for audit purposes, and each entry is attributed to the data source, including a full literature reference, the source URL, and a DOI (where available). This ensures that we can display this data interactively and that the original source remains permanently available. We also included any digital identifiers from vector datasets so that the individual feature can be tracked and mapped over subsequent data revisions.

We linked each entry into openDELvE to a delta using the PolygonID from Edmonds et al., (2020), and additionally flagged deltas that need manual review in the future. It ensures that there is a robust process in the future to signal amendments needed or entries for which sources are undocumented or inaccessible. This supports maintenance and prevents repetition of previous search activities.

**2.6 Data management**

The resulting data layers for levee area and levee line feature were created in ArcGIS Pro and hosted on an ArcGIS Online data hub (http://www.opendelve.eu, Figure 3). Additionally, we maintained ongoing research data exports in the DataverseNL environment as the database develops, which also assigns permanent identifiers (DOIs) to the research dataset. Data is stored in three defined entities as per Table 1, and we stored each layer within their own container in the public ArcGIS Online® environment. These layers were then published to be used as part of the ArcGIS Online Directory and through modern GIS clients via a Web Feature Service (WFS).

The openDELvE platform facilitates an interactive and community driven maintenance of the dataset through an amendment form and additional messages in all metadata files. Suitable new data will be added to openDELvE by the authors at Utrecht University, and made public on the openDELvE webpage and the DataverseNL environment.

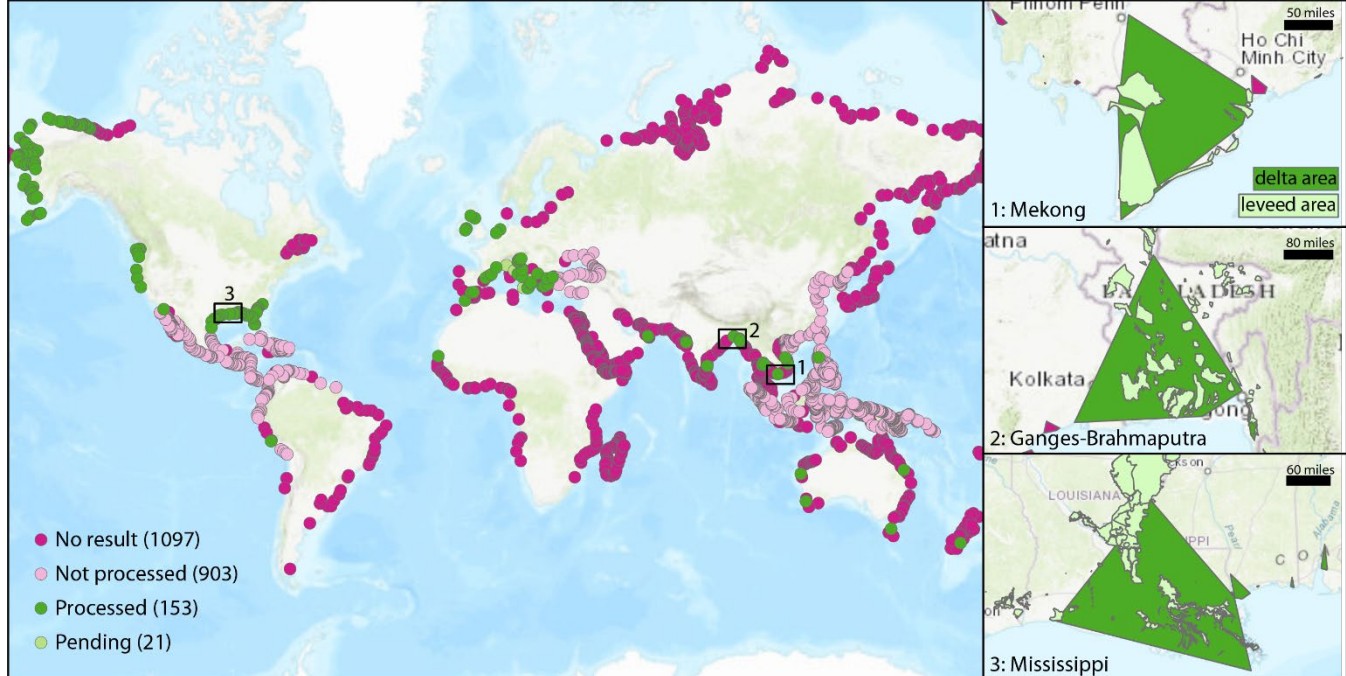

**Figure 2: Distribution of delta levee dataset completeness and data availability in release of openDELvE (v1.0). Polygons encompass the four-point deltaic extent as defined by Edmonds et al. (2020).**

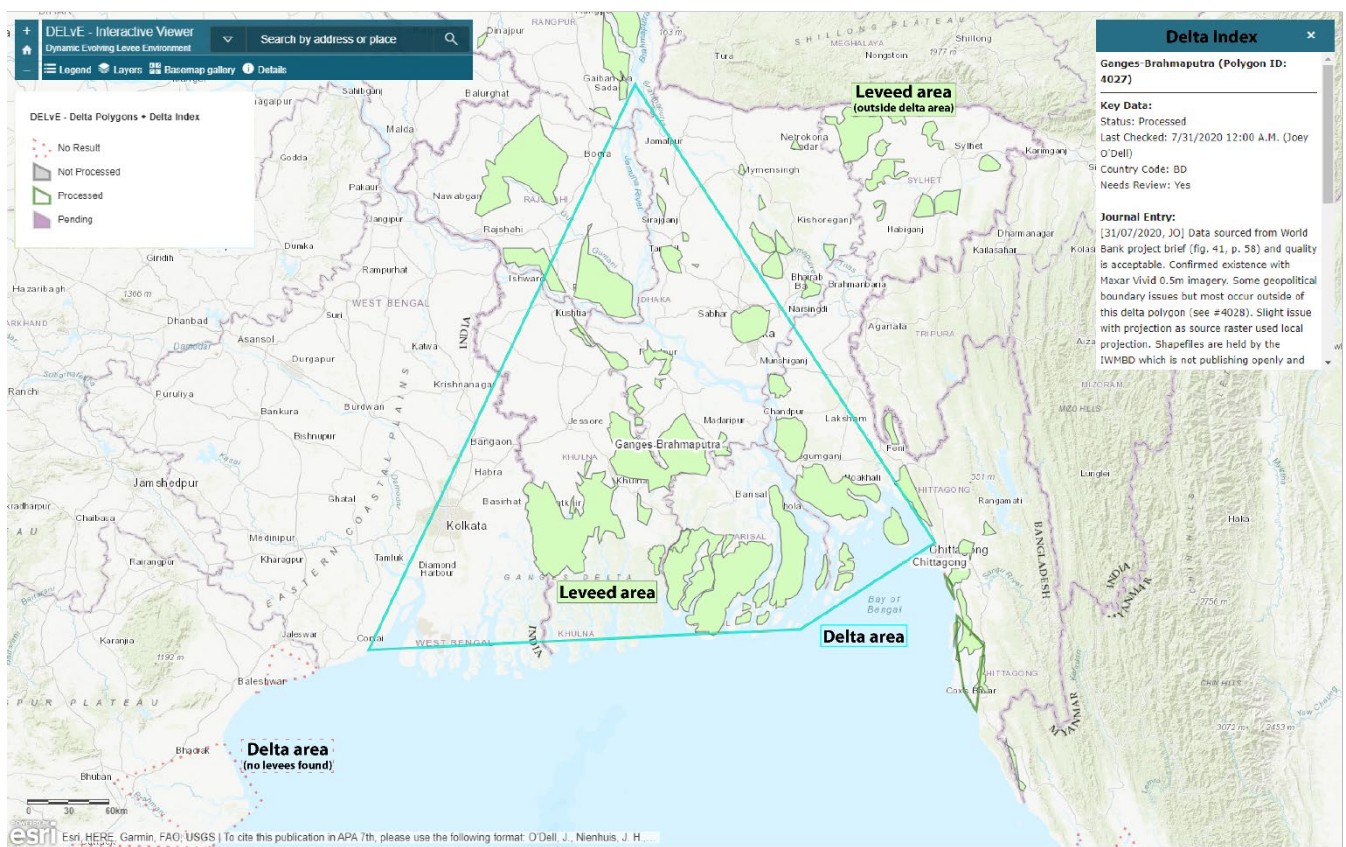

255 **Figure 3: Interactive browsing interface to openDELvE built using the ArcGIS Online platform. Area of focus is the Ganges-Brahmaputra delta, Bangladesh. Data content as per openDELvE version 1.0. Available publicly at: http://opendelve.eu**

## 2.7 Data assurance

Before releasing the dataset, we performed several checks on the data and metadata (Table 4). We then generated metadata compliant with the EU INSPIRE geospatial metadata standard (European Parliament, 2007) using the built-in ArcGIS® Pro

260     wizard for each data element (Table 2), and for the dataset in its entirety. This included interactive help-text for the model builder. We checked the metadata files for completeness using the metadata wizard in the ArcGIS® Pro system.

**Table 4: Categories and criteria for the data validation performed on the dataset**

| Type | Criteria |
|---|---|
| Duplicate Check | There are no duplicate delta polygon IDs (*PolygonID)* in the index |
| Orphan Check | All linked delta polygon IDs matched a delta polygon in the dataset<br>There were no unsuccessful joins between the data layers |
| Null Check | Where there was no match to a delta polygon, this returned -1<br>Where it was not (yet) possible to match the polygon to a delta, this returned null |
| Visual Check | Visually verify if data appears reasonable (i.e., within 100 km of delta polygon border, within proximity of water feature, of a shape that is coincident to delta morphology) |
| Metadata Check | All fields in the ArcGIS® Pro metadata wizard completed |

### 2.8 Applications of openDELvE

#### 2.8.1 Land-use assessment with Copernicus Global Land Cover Layers

We used the Copernicus global land cover dataset (Buchhorn et al., 2020) to identify land use types and patterns within deltas and within leveed areas on deltas. Copernicus land cover data separates 16 natural vegetation classes, 4 non vegetated classes, and 2 human-influenced land cover classes, on a global 100-m grid. We selected the land cover data from 2019 and calculated for each land use and for each delta the area that is either protected from or exposed to flooding.

#### 2.8.2. Population density with LandScan™

The Oak Ridge National Laboratory's LandScan™ population data (https://landscan.ornl.gov/) was used to calculate population density within levees and outside of levees. LandScan provides globally yearly gridded data at a 30-arc sec (~1 km) resolution, counting resident and transitory population. We used data from 2020 and calculate delta population within and outside leveed areas for each processed delta polygon.

#### 2.8.3 Coastal flooding analysis with COAST-RP

The COAST-RP dataset (COastal dAtaset of Storm Tide Return Periods) of Dullaart et al. (2021) provides spatial extent of coastal floods from storms at 30-arcsec resolution for storm return periods from 1 to 1000 years. This is based on a global hydrodynamic model of the ocean that provides coastal water levels. COAST-RP then propagates these water levels in land using a static inundation model on top of a state-of-the-art global elevation dataset and assuming a water level attenuation factor based on distance. COAST-RP does not include levee data. Here we intersected openDELvE data with COAST-RP to estimate simulated flood extents that might, in reality, be protected from coastal flooding by levees. We assessed storm return periods of 10, 100, and a 1000 years, but, because of limited levee height data and levee quality data, we have not assessed actual protection but rather potential protection.

### 3. Results

#### 3.1 openDELvE extent & summary

The current release of openDELvE contains 11,188 levees with a combined length of 19,248 km. These levees protect 1,657 separate areas that collectively span 44,734 km², of which 41,399 km² are on a delta (following definition in Sect. 2.4) (Table

5). Most of the data in openDELvE (97% of the leveed area) is derived from vector or high-resolution raster sources and is of good quality (Figure 4). We have processed levee information for 153 of the 2,174 deltas identified by Caldwell et al. (2019), representing 28% of the global delta area (246,885 km$^2$ of 874,142 km$^2$) (Figure 2). Another 1,097 deltas (59% of global delta area) are pristine. Levees are unlikely and we did not find information on levee presence, nor we could we identify levees visually (No Result category). This includes deltas like the Amazon and Lena. A further 924 deltas remain unprocessed, largely because data is unavailable —these are also small and collectively represent 12% of global delta area. We have processed the largest deltas and the remaining deltas are less likely to have levees.

Levees protect 17% (41,399 km$^2$) of delta area for the 153 deltas included in openDELvE, but protection varies regionally. It is 2% in Asia-Pacific but 39% in Europe and C. Asia, and this broadly reflects levee presence but also data availability and data publishing policies between different regions (Figure 4). Protected delta area also varies per delta, from fully unprotected deltas such as the Colville (0%, USA) to mostly protected deltas such as the Rhine-Meuse (70%, NL). Our delta areas also include (coastal) surface water, which is 20% of the Rhine-Meuse land area, therefore the protection percentages will be higher if only land is considered.

| Continental Zone (UN Region) | Number of deltas with levee data present in openDELvE[a] | Total number of unique leveed areas represented in dataset | Total deltaic area[a] [km$^2$] | Area protected by levees within the delta[a] [km$^2$] | Coverage of delta area by levees as % of deltaic area (computed) |
|---|---|---|---|---|---|
| Africa | 3 | 9 | 4,352 | 569 | 13% |
| Americas | 100 | 301 | 105,766 | 13,000 | 12% |
| Asia-Pacific | 19 | 83 | 130,000 | 2,110 | 2% |
| Europe & C. Asia | 31 | 86 | 6,492 | 2,560 | 39% |
| **Processed Total** | **153** | **479** | **246,885** | **41,399** | **17%** |
| No Result | 1,097 | - | 519,039 | - | - |
| Unprocessed (Pending & Not Processed) | 924 | - | 108,218 | - | - |
| **Global Total** | **2,174** | **479[a]** | **874,142** | **41,399** | **5%** |

Table 5: Summary of processed features and deltaic area at openDELvE (current release) per geographic region and area totalled, rounded to nearest integer.

[a]openDELvE contains 1,601 leveed area polygons but they are partially overlapping due to the structure of administrative units in the USACE NLD. Overlapping sections are only counted once for the purpose of this table.

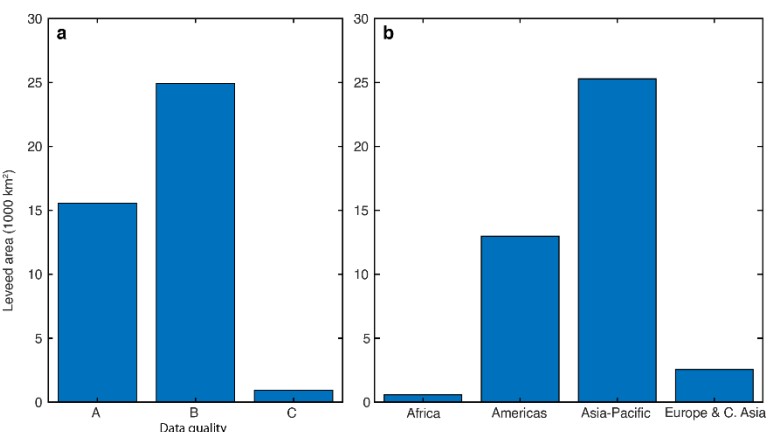

Figure 4: (a) Distribution of data quality classification in openDELvE given for each individual leveed area feature, classified according to the data quality matrix (Table 3). (b) Distribution of leveed area data in openDELvE by UN Region.

## 3.2 Demonstrative applications of openDELvE

Data on levees can bring important insights and more accurate predictions in delta studies. Levees are sometimes included in small-scale studies, but not yet in large-scale or global studies (e.g. Dullaart et al., 2021, Nienhuis and van de Wal., 2021). Global studies are becoming more common, in part because of global challenges such as climate change (IPCC, 2021).

315

Here we showcase uses of openDELvE, including flood-protection of land use (what type of land is protected and what will be at risk), flood-protection of delta population (how many people live in flood-protected vs flood-prone areas), and potential improvements of flood hazard models in deltas (Figure 5).

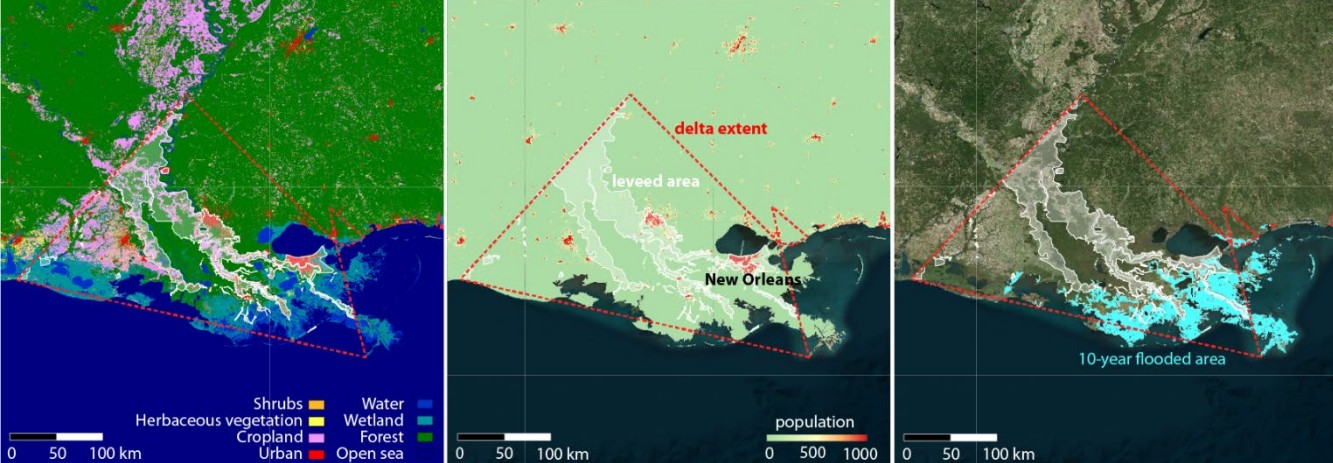

320 **Figure 5: Examples of land use (Copernicus Global Land Service, Buchhorn et al., 2020), population (LandScan, https://landscan.ornl.gov/), and flooded area (Dullaart et al., 2021) within and outside levees in the Mississippi Delta**

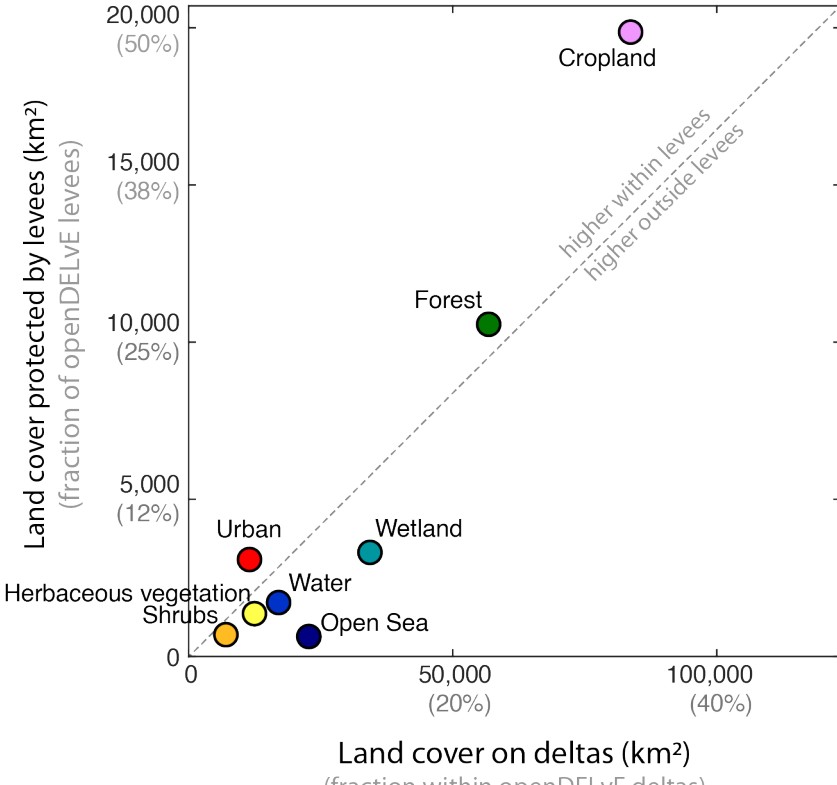

**Figure 6: Average land cover for 153 deltas within flood-protection levees compared to deltas as a whole.**

First, an intersection between openDELvE and land use data shows that land-use patterns differ significantly inside leveed areas compared to the rest of the delta (Figure 5, Figure 6). Urban and built-up land are concentrated within leveed areas, 325 whereas wetlands and water bodies are more likely to be found outside levees. For example, 48% of flood-protected delta area

is used as cropland, compared to 31% of the non-flood-protected delta area (Figure 6). Over- and under representation of different land use classes is likely because levees are constructed to protect land with higher value, such as urban, built-up areas and croplands. Levees are therefore important for food availability and access (Islam and Al Mamun, 2020), protection

330    of urban centres and urban infrastructure (Jongman et al., 2012), and for reducing exposure to flooding (Lumbroso et al., 2017). There is a second effect that can also play a role. The existence of levees could lead to greater investment and development of urban and agricultural land compared to areas outside levees (Hutton et al., 2019), the so-called "levee effect". openDELvE does not include year of construction for levees, so that it is not possible to separate these two effects.

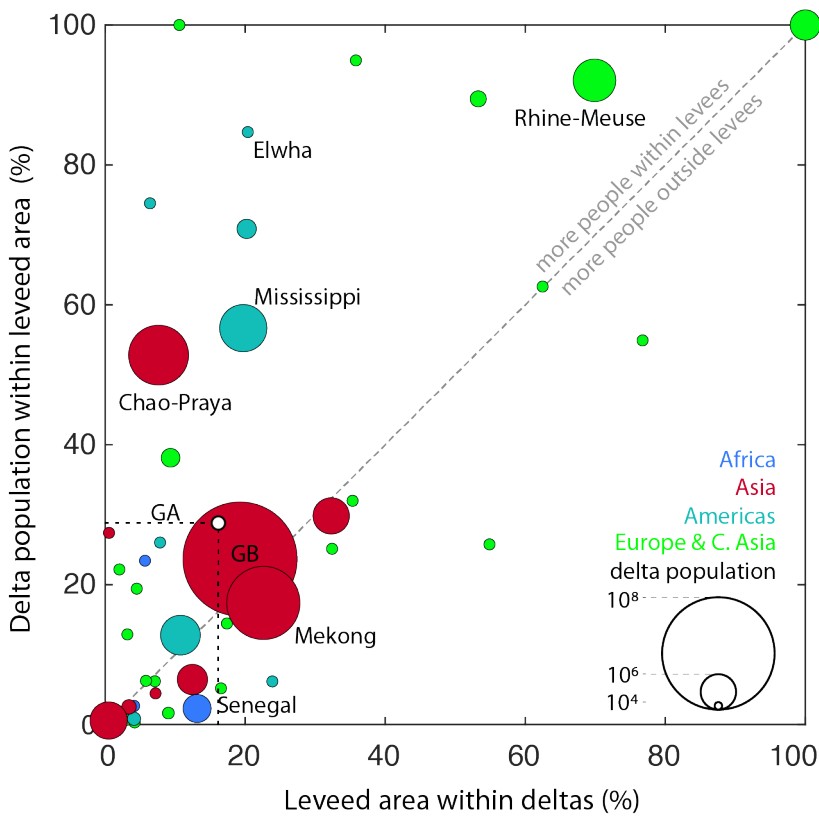

**Figure 7: Flood-protected delta area vs flood-protected delta population, both as a fraction of the delta total. GB is the Ganges-Brahmaputra. The dotted line indicates the global average (GA).**

Second, our analysis with openDELvE and population data suggests that, for the 153 deltas in openDELvE, 74% of delta population lives outside flood-protected areas. Population densities are higher inside flood-protected areas: leveed areas occupy 17% of delta area, on average, but protect 26% of delta inhabitants. However, these global averages hide large

340    differences between deltas and regions (Figure 7). In Europe (85%) and the Americas (41%) we find a large fraction of the delta population to be protected, e.g., the Rhine-Meuse delta in the Netherlands (92%), and the Mississippi in the USA (57%, Figure 5). This is not the case across Africa (3%) and Asia-Pacific (24%). Looking at population densities the pattern is different. In Asian deltas, 800 people per km$^2$ live outside flood-protected areas, compared to 15,000 people per km$^2$ inside. In contrast, in Europe and the Americas there is only a 5 and 9 fold increase in population densities within flood-protected

345    areas, respectively. The different patterns could be the result of competing factors in the co-evolution of levees and cities. Although levees are constructed to protect people and are therefore expected to protect the most populated areas, they are also constructed in vulnerable delta locations, away from likely locations of major cities in regions that historically did not build levees (e.g., Bangladesh).

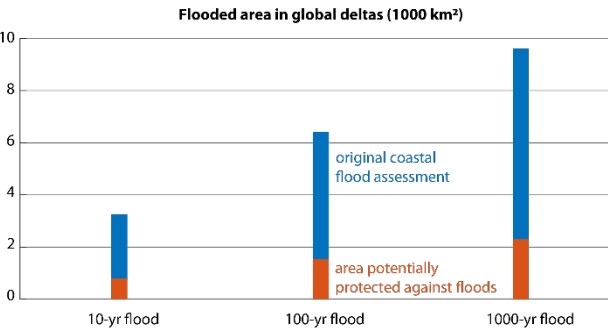
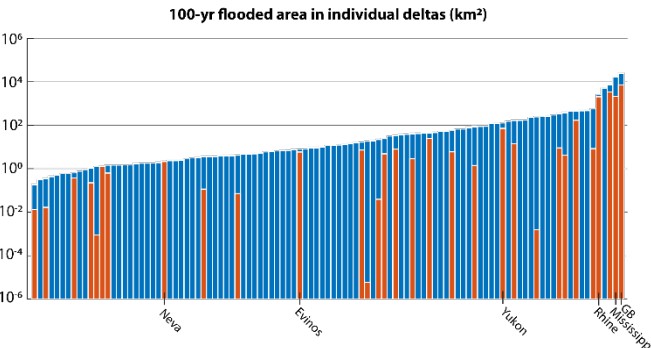

**Figure 8: Delta area potentially protected against coastal floods (in red), compared to all exposed delta area (in blue), for all deltas (left) and individual deltas (right). GB in the Ganges-Brahmaputra.**

In a third demonstration of openDELvE use, we assess the intersection of levees with global coastal flood assessments (Figure 5, Dullaart et al., 2021). When neglecting the presence of levees it would seem that 13% (32,261 km$^2$) of the combined area of the 153 deltas is exposed to coastal floods with return period of 10 years (Figure 8). This increases to 26% (63,179 km$^2$) and 39% (95,879 km$^2$) for 100-year and 1000-year floods, respectively. However, when accounting for levees in openDELvE, we find that these could reduce flood exposure by 25% (8,206 km$^2$) in the case of 10-year floods, and by 24% (22,744 km$^2$) in the case of 1000-yr floods (Figure 8). Protection against floods varies greatly between deltas. For the Rhine delta it is 78%, in the Ganges-Brahmaputra delta it is 29%, and in the Mississippi delta it is 13% (Figure 5, Figure 8). Since openDELvE does not include data on levee heights and levee protection standards, we cannot associate each levee with a magnitude of flood; therefore, these numbers represent an approximation of the best-case protection offered by levees.

## 4.0 Discussion

### 4.1 How representative is openDELvE?

As summarised in Table 5, we found that 17% of the delta area processed in openDELvE is protected by levees. This should be considered a rough estimate and it is difficult to assign a global uncertainty. Delta area is notoriously difficult to define: data on delta area from two studies (Edmonds et al., 2020, Syvitksi & Saito, 2007) vary by 30%, on average, per delta. For levees registered      by nationally maintained databases (e.g., Mississippi, Rhine-Meuse) the data quality is good. There, there is rich metadata and a lower (but no zero, see Knox et al, 2022)      chance of false negatives (openDELvE missing existing levees). Data quality and coverage in other deltas (e.g. Ganges-Brahmaputra, Mekong) is poorer, and this appears to be linked to the lack of a nationally or regionally coordinated platform for levee data sharing. There, the chance is higher of false negatives and      undercounting of leveed area.

Trying to assess global leveed area for all 2,174 global deltas, including the unprocessed and no result categories, the fraction of delta area that is flood-protected is likely to be lower than 17%. Many of the "No Result" deltas are in sparsely populated areas (the Amazon, the Arctic). We expect those to have fewer levees compared to the 153 deltas within openDELvE. Global delta levee area is probably higher than 5%, given that this would mean openDELvE currently includes all levees on deltas. The fraction of delta area that is protected can also be somewhat greater than 17% because of limited levee data availability in openDELvE, in Asia in particular.

### 4.    2 Global barriers to data availability

Data sovereignty is an emerging topic within global modelling that revolves around the value, sharing, and ownership of data in a global context. Whilst we acknowledge that breakthroughs have been made in the academic world of data sharing, through the formation of data initiatives (e.g., FAIR) and for standardised data sharing (e.g., INSPIRE, European Parliament, 2007),

data in the private and governmental sectors can still be considered as an internal asset. Tang et al. (2020) define the term 'data sovereign' to identify someone with the capabilities, skill set, and hierarchical position to facilitate data sharing across borders.

In our search for information, we realized that that countries and governmental organisations which have core values supporting open data tend to treat levee information as a 'product' and therefore appoint a central data repository or facilitated ordering process to act as a 'data sovereign'. Some repositories may not themselves hold the actual data but act as centrally maintained indices of national data. Examples of national repositories are the US data.gov platform, which holds record locators for the US Army Corps of Engineers *National Levee Database*, the UK data.gov.uk Open Data platform, which holds record locators for the UK Environment Agency *Asset Information Management System*, the Dutch data.overheid.nl, which holds record locators for the Rijkswaterstaat (Dutch Ministry of Infrastructure and Water Management) *Dataregister*, and Australian data.gov.au, which holds record locators for the various state-led systems in place across the country.

Other countries and institutions treat data differently, which can act as a roadblock to progress towards a harmonised global database. We found fewer data for deltas in Africa, China, South-East Asia, the Southern and Central Americas - as well as those in the Russian Federation and late-accession members to the EU. Data is often stored in archives that we were unable to access, and as such this is only a partially complete dataset.

### 4.3 Uncertainties in openDELvE applications

Our example applications using openDELvE data are uncertain. Global data resolutions vary, which can result in inaccuracies when overlaying grids. Land cover (Buchhorn et al., 2020), population (https://landscan.ornl.gov/), and coastal flooding (Dullaart et al., 2021) data are available at 100 m, 1 km, and 1 km, respectively. Vectorized leveed area data is generally available at a higher resolution (~1 m, e.g., levees.sec.usace.army.mil) and can therefore dissect coarser gridded data. In addition, there are other uncertainties inherent to the source itself that need to be taken into consideration when using the data, but some remain unquantified. For example, the coastal flood maps that we use (Dullaart et al., 2021) have not been validated because flood observation data remains too sparse for a detailed analysis of the uncertainty.

These data uncertainties affect our results. The 26% reduction of the 100-yr flood exposure by delta levees that we report (section. 3.2) could be lower or higher. The reduction could be higher because we miss levees in openDELvE (Knox et al., 2022), but it could also be lower because the delta area is poorly defined. Land cover and population statistics (section 3.2) could be similarly affected by data uncertainties. The large, reported fraction of delta population that is protected (e.g., Europe: 85%, Americas: 41%) could be even higher. Some of the remaining population could accidentally be included but not live on the delta proper and therefore not reside in a flood zone. It could be also lower because some population might be outside the delta area and outside the leveed area. Quantifying these uncertainties is challenging.

### 4.4 Future outlook

By publishing our data as openly as possible (following FAIR principles) we intend to encourage not only external inspection but also suggestion of changes and further data additions. For this reason, we developed a webpage ([www.opendelve.eu](www.opendelve.eu)) and a new-data submission system. We encourage users to refer us to levee data that we missed, and seek partnerships with local experts of countries for additional data inclusion. Additional crowdsourced or "*volunteer geographic information" (VGI,* Young et al., 2020) projects such as 510 - an initiative of the Netherlands Red Cross - and OpenStreetMap.org may be able to further expand data on levees. We recognise the work of Young et al. (2020) in documenting the deployment of, and challenges associated with, a globally diverse data collection project, however including crowed-sourced data was out of the scope of our research.

Levee data can also be expanded using different means. There is the possibility of openDELvE to function as a training dataset for statistical (machine learning) models for levee and flood detection (Wing et al., 2019). By publishing our data with an open licence (Creative Commons Attribution) we encourage its reworking and reuse.

## 5. Conclusion

openDELvE is a global delta levee database. We have standardised levee attributes and features from disparate data sources
to allow for global comparability and obtained a database of 11,188 levees with a combined length of 19,248 km. For the deltas in openDELvE we find that 41,399 km$^2$ of their area is contained within levees. This represents 17% of their area and 5% of global delta area. Levees predominantly protect delta cropland, which comprises 48% of protected delta area. Only 26% of delta population is protected by levees, but this varies greatly across deltas, from 3% in some deltas in Africa to 92% in the Rhine delta. Levees potentially protect up to 8,206 km$^2$ (10-year floods) or 22,744 km$^2$ (1000-yr floods) of delta land against
flooding.

openDELvE can improve delta flood hazard modelling, global delta hazard assessment, and studies of sustainable delta management in the face of sea-level rise and other anthropogenic pressures. Our database is biased due to data availability, with more data available for Europe, Central Asia, and the Americas than for Africa and Asia-Pacific. openDELvE is FAIR,
openly available, and we encourage contributions from other researchers and experts via http://www.opendelve.eu.

## Code availability

The ArcGIS® Model Builder template used to process vector data is published within the research dataset, available on DataverseNL at https://doi.org/10.34894/2WZ0S9.

## Data availability

The research dataset is publicly available on DataverseNL at https://doi.org/10.34894/2WZ0S9.
The layers and viewing interface are publicly consultable at http://www.opendelve.eu and are additionally hosted in the ArcGIS Online Portal for use with ArcGIS® and other OGC-compatible GIS packages. Additional data used in the applications section (population, land-use, flooded area) are available through original sources, with findings per delta summarized in Supplementary Table 1.

**Author contribution**

JO curated and maintained the data, performed the investigation, and prepared the draft manuscript with contributions from the co-authors. JHN conceptualised, validated, and supervised the project, visualised the results, and edited the manuscript. JRC supervised the project, visualised the results and edited the manuscript. DAE and PS provided key digital resources and edited the manuscript.

**Competing interests**

The authors declare that they have no conflict of interest.

**Disclaimer**

Figure 1, 2 & Supplementary Figure 1 were created using ArcGIS® software by Esri (ArcGIS® Pro [ver. 2.6.2] and ArcGIS®
Online). Figures 4-8 were created using Matlab 2021a. Cartographic data displayed in the images is supplied by Esri under
licence from HERE, Garmin, FAO, NOAA, EPA, NPS, and the USGS.

**Acknowledgements**

The authors wish to thank the many organisations who have actively chosen to embrace open data standards and share such
rich datasets as this is a crucial part of the database. We are grateful to Ece Özer (TNO) and Alex Curran (formerly of TU
Delft) whose work on the SAFElevee project and the ILPD, and their detailed publications as a result, were instrumental in
the discovery of closed datasets. We further thank Silvia Barbetta, Albert Kettner, Yoshiki Saito, and Dhruvesh Patel for their
help in gaining access to international data. We would additionally like to thank Fergus Miller Kerins for his help in testing
and exploring deployment possibilities for the dataset. We thank Job Dullaart and Sanne Muis for providing access to the
COAST-RP flood return dataset. Finally, we are very appreciative of the support from Maarten Zeylmans van Emmichoven
that allowed the data processing to continue remotely during the pandemic. JO is supported by funding from the WCFD at
UU. JHN is supported by NWO vi.veni.192.132. JRC is supported by Rivers2morrow.

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
