# Peer review of "A global open-source database of flood-protection levees on river deltas (openDELvE)"

_Natural Hazards and Earth System Sciences, 2021_

## Author Response (AR1)

Reply to the editor and reviewers (in *italics*), line numbers refer to the tracked-changes manuscript text below.

**Editor Lindsay Beevers**

Dear Joey O'Dell et. al.

Thank you for the submission of your manuscript " A global open-source database of flood-protection levees on river deltas (openDELvE)". In general this is a well written paper reporting on a potentially important dataset, which could be of great interest to the community.

As you know, two reviewers have now provided thoughtful reviews, which you have replied to online. These reviewers highlighted some questions around the suitability of the manuscript for this journal. Based on these queries the Editorial team have had a detailed discussion. In summary we believe that this paper can be a useful and interesting addition for NHESS (and it is not without precedent for NHESS). However, for the paper to be publishable further analysis of the data presented is necessary.

Therefore I would like to invite you to submit a revised version of your manuscript. In particular, we are keen to see an elaboration of the data, some further analysis and some discussion of the trends observed.

Please concentrate on expanding the objective to include:

a. explicit connection of the database to the natural hazard agenda

> *Thank you for your positive comments and the opportunity to provide a revised manuscript.*
>
> *We have made substantial improvements to the manuscript, and now include an explicit connection to the natural hazard agenda. For example, we have analyzed global coastal flood hazard simulations (which do not yet include levee data), and calculated how much of the delta flood exposed land is potentially protected by levees. We find that this flood hazard model exaggerates delta flood exposure by 33% on average, but 100% for some deltas (shown in new figures 5 and 8).*
>
> *L166: "We find that current flood hazard assessments may exaggerate the delta flood exposure by 33% on average, but up to 100% for some deltas."*
>
> *We also find that 74% of delta population lives outside flood-protected areas (L649) and that:*
>
> *L849: "Levees potentially protect up to 8,206 $km^2$ (10-year floods) or 22,744 $km^2$ (1000-yr floods) of delta land against flooding."*

b. as RC1 requests - further exploration of 'leveed areas' and how these are computed, and an analysis and discussion of who/what they protect

*We have now substantially expanded our explanation of "leveed areas", and also analysed land use and population that is potentially protected by levees.*

*Our new figure 1 shows how we convert levee information to leveed area estimates (L449). We discuss the analysis in a new section (2.4.3) and also in the supplementary methods (S3).*

*Our analysis of land cover within and outside levees in deltas shows that:*

*L613: "First, an intersection between openDELvE and land use data shows that land-use patterns differ significantly inside leveed areas compared to the rest of the delta (Figure 5, Figure 6). Urban and built-up land are concentrated within leveed areas, whereas wetlands and water bodies are more likely to be found outside levees. For example, 48% of flood-protected delta area is used as cropland, compared to 31% of the non-flood-protected delta area (Figure 6)."*

*In another analysis where we use the LandScan global population data, we find that:*

*L652: "leveed areas occupy 17% of delta area, on average, but protect 26% of delta inhabitants. However, these global averages hide large differences between deltas and regions (Figure 7). In Europe (85%) and the Americas (41%) we find a large fraction of the delta population to be protected, e.g., the Rhine-Meuse delta in the Netherlands (92%), and the Mississippi in the USA (57%, Figure 5). This is not the case across Africa (3%) and Asia-Pacific (24%). Looking at population densities the pattern is different. In Asian deltas, 800 people per km$^2$ live outside flood-protected areas, compared to 15,000 people per km$^2$ inside. In contrast, in Europe and the Americas there is only a 5 and 9 fold increase in population densities within flood-protected areas, respectively."*

c. some more analysis and discussion of the trends in the data and what this may mean within the context of the journal (this may reflect back to point a above).

*We have substantially expanded our "results" section, beyond the statistics we provided earlier. Figure 5, 6, 7, and 8 are new and show how our levee dataset can be used to provide insights into flood protection on deltas (what is protected, who is protected), but also for improvements in hazard modelling (what is exposed).*

*We believe these new analyses have generated key insights into flood-protection on deltas, which fit well in the context of the NHESS: There is a substantial (25%) reduction in global coastal flood exposure estimated by flood models with vs. without levees. Yet, our data shows that 76% of delta population lives outside levees and are not protected at all.*

Would you please also provide an 'author's reply' to the this and the reviewers comments (feel free to use the same words that you used in what you have already uploaded and elaborate on this with the added analysis) and include, in the same author reply document, a track changes document between the old manuscript and the new one.

I look forward to seeing the next version of your manuscript and will send it out again to the previous reviewers.

*Thank you. We have added additional responses to the reviewers, and have also included a track-changes version of the manuscript below.*

*Thank you for your consideration,*

*Jaap Nienhuis, also on behalf of co-authors Jana Cox, Joey O'Dell, Douglas Edmonds, and Paolo Scussolini*

**RC1**

The authors present an important and useful database of flood protection structures in levees globally. An assimilation of such data is crucial for improved understanding of global flood risk, yet is one of the few topics in our field which has not undergone meaningful recent advances in its characterisation (at least, in an efficient or somewhat automated way). While not wanting to understate the importance of the task the authors have completed, the presented science available to review is extremely minimal. It is paradoxical that the authors' work is more meaningful and important than much of the marginal advances typically presented in the modern deluge of academic papers, but the paper in its present form does not appear to be publishable in this kind of journal.

The decision is really, then, an editorial one. As a reviewer, I can not see why this wasn't simply submitted to a dedicated journal that fields papers describing datasets. If the authors want to publish in a traditional journal, they need to actually apply the data they have collected to answer a research question. One idea would be to look at recent European or global flood modelling studies by the JRC and examine how their risk estimations change when considering the information in this database.

*We thank the reviewer for their review, and for their positive words about our paper. We decided to submit to NHESS because we felt the community and readership would be very interested to learn about our new research and our derived global finding on river deltas: of the deltas in our database, we find that 17% of their habitable area is confined within flood-protection levees. At the same time, we decided against including substantial new derived analyses in this manuscript because we felt it would mask the major contribution that is the*

*openDelve database itself. Instead, we provide database statistics and extensive methodology and focus the manuscript on a single message: openDelve.*

*Our decision to submit to NHESS was also inspired by another recent and well-read NHESS study on a flood protection database FLOPROS (https://doi.org/10.5194/nhess-16-1049-2016), by one of our co-authors. We fully agree that a decision about our manuscript at NHESS is ultimately then an editorial one.*

*In response to the editorial decision, we have now added 3 major analyses that improve our understanding of flood-risk on deltas.*

*For example, we estimate based on our data that 74% of delta population lives outside flood-protection levees (Fig. 5 and 7) and that levees primarily protect cropland (Fig. 6). Existing flood-hazard assessments could exaggerate delta flood exposure by 33%, even for (modest) 10-year flood event, because they do not (yet) include levees (Fig. 8).*

I would also like to see the delineation of a "leveed area" unpacked further. How is this defined? If we know the location of levees, it is not straightforward to understand who they actually protect (presumably some kind of "undefended" model would be needed). If conclusions are to be drawn based upon these areas, the authors should provide more detail on how they are computed.

*Regarding the leveed area, we agree that this is an important point and we have added more explanation on how this was derived. Figure 1 gives a graphical overview of how we derive levee area from levees, and is further explained in the new section 2.4.3. The process is also explained in the supplementary Fig. 3.*

**RC2 – Hamed Moftakhari**

Here the uthors have developed and presented a very useful open-source global database that provides information on river delta levees and the area protected by them from flooding. While the database is interesting and the manuscript is well-written, I am afraid if I see any explicit research question to be explored by the authors. I mean, while the database can be beneficial for modeling and flood hazard assessment and would provide an excellnt ingredient for research projects, in this submission the scintific contribution and novelty in missing.

So, my suggestion is that authors reconsider the outlet, and consider other venues/journals (i.e. Scientifc Data https://www.nature.com/sdata/) that better suite the scope of this draft. However, I'd leave it to the editor and authors to decide.

Also, on a very minor suggestion, I encourage to enrich your discussion in "1.2. Why levees matter" to explain the concept of levee effect and how it has been contributed to the increased exposure to flooding in the past. See this for example (https://www.pnas.org/content/103/40/14653)

*We thank Hamed Moftakhari for his review and for his positive words about openDelve. Some of the comments were also expressed by reviewer #1. To them, we responded that we submitted our manuscript to NHESS because we felt the community and readership would be very interested to learn about our new research and our derived global statistics on river deltas: of the deltas in our database, we find that 17% of their habitable area is confined within flood-protection levees.*

*We thank Hamed also for pointing us to this interesting work on the "levee effect". We have included this in our revision because of its relevance for the paper. Four of us live behind levees in the Netherlands. It is even relevant for us.*

*We discuss the levee effect (in L227), and also point out how our data could be used (in the future) to assess the levee effect (L624). We would need a time of construction of levees (available for some levees within openDELvE, primarily those for the USACE national levee dataset), and timeseries of land use or population datasets. Jointly, we can then ask what came first: the development that then needed protection, or the protection that then attracted development.*

[revised manuscript text omitted]

~~Levees themselves are not new creations, and so the majority of data that references their locations and standards is historical and locked away in paper form (maps, plans etc.). However their effect on the human population can be easily seen. Modern urbanized deltas tend to be heavily embanked by levees because high population densities have demanded protection against river and coastal flooding. Despite these levees, people living in coastal deltas face mounting threats:; they are disproportionately affected by coastal flooding and relative sea level rise (Edmonds et al., 2020) and rely on a river sediment supply that is ever diminishing river sediment supply (Dunn et al., 2019). Data on levees can help to assess these threats.~~

~~This is then further compounded by the "levee effect", defined by Gilbert White in 1947 whereby increased levee building creates a false sense of security which can often come at the effect of land use decisions (Hutton et al., 2019) and is particularly well documented in New Orelans both in the lead up to Hurricane Katrina and in the multiple waves of reconstruction post-disaster (Kates et al., 2006). Therefore the demand for levees receives a certain public sentiment through the apparent afforded protection and not the actual standard of protection offered in the area.(Hersher, 2018).~~

[revised manuscript text omitted]
 ~~processed in openDELvE (~ 150 deltas)at awithhow much delta areadifferences in land use occurrence betweenandfromdelta landThe MODIS raster data was loaded into ARCGIS Pro for anaylsis. Using Zonal Statistics, the majority land use within and outside the leveed areas was identified within each processed delta polygon. The percentage of all land use types was calculated using Tabulate Area.~~

**2.8.2. Population density with LandScan™**

The Oak Ridge National Laboratory's LandScan™ population data (https://landscan.ornl.gov/) was used to  calculate population density within levees and  outside of un . LandScan provides  globally yearly gridded data  at a  30-arc sec (~1 km) resolution, counting averaged over 24 hours, therefore not just counting  and transitory population . We used data from 2020 and calculate delta population within and outside leveed areas  for each processed delta polygon ~~of openDELvE (~ 150 deltas) using Zonal Statistics~~.

**2.8.3 Coastal flooding analysis with COAST-RP**

The COAST-RP dataset (COastal dAtaset of Storm Tide Return Periods) of Dullaart et al. (2021) provides spatial extent of coastal floods from storms at 30-arcsec resolution for storm return periods from 1 to 1000 years. This is based on a global hydrodynamic model of the ocean that provides coastal water levels. COAST-RP then propagates these water levels in land using a static inundation model on top of a state-of-the-art global elevation dataset and assuming a water level attenuation factor based on distance. COAST-RP does not include levee data. Here we intersected openDELvE data with COAST-RP to estimate simulated flood extents that might, in reality, be protected from coastal flooding by levees.  We assessed storm return periods of 10, 100, and a 1000 years, but, because of limited levee height data and levee quality data, we have not assessed actual protection but rather potential protection. and~~levees for of various return periods in deltas. This dataset is combined with openDELvE to indicate how the presence of levees can alter the prediction of coastal flooding. The spatial extent of floods for 10, 100 and 1000 year return periods was calculated for the processed delta polygons of openDELvE (~ 150 deltas) and the leveed areas using Tabulate Area.~~

**3. Results**

**3.1 openDELvE extent & summary**

The current release of openDELvE contains 11,188 levees with a combined length of 19,248 km. These levees protect 1,657
separate areas that collectively span 44,734 km²  of which 41,399 km² are on a delta (following definition in Sect. 2.4) (Table 5). Most of the data in openDELvE (97% of the leveed area) is derived from vector or high-resolution raster sources and is of good quality (Figure 4).  We have processed levee information for 15  of the 2,174 deltas identified by Caldwell et al. (2019), representing 5% of
the global delta area (244266,885 km² of  874,142 km²) (Figure 2) We find 1,097 deltas (59% of global delta area) are pristine  where Levees are unlikely and  we did no find information on levee presence , nor we could not identify levees visually  (No Result category) 22% . This
includes deltas like the Amazon and Lena. A further 92 deltas remain unprocessed, largely because data is unavailable — these are also small and collectively represent 12% of global   delta area We have processed the largest deltas and the remaining deltas are less likely to have levees.

Levees protect 17% (41,399 km²) of delta area for the 153 deltas included in openDELvE, but

Considering only processed deltas, 1910% (44,73443,875 km² of 239,044426,663 km²)  protection varies regionally.
It is 2% in Asia-Pacific but 39% in Europe and C. Asia, and this broadly reflects levee presence but also data availability and
data publishing policies between different regions (Figure 4). Protected delta area also varies per delta, from fully unprotected deltas such as the Colville (0%, USA) to mostly protected deltas such as the Rhine-Meuse (70%, NL). Our delta areas also include (coastal) surface water, which is 20% of the Rhine-Meuse land area, therefore the protection percentages will be higher if only land is considered.

~~Total global minimum delta area extent is 847,936564,771 km² (Edmonds et al., 2020), which means that, at least, 58% of global delta area is within verifiable levees. This number should be considered a minimum as many deltas remain unprocessed and we suspect that many levees exist that are not (yet) in the openDELvE dataset as even when discovering data, there existed data sources that were incompatible with the licensing of the dataset, for which we document in the delta index.~~

~~Percentage coverage of delta area by levees between continentals zones (using the UN Region from Edmonds et. al. (2020) ranges from 13% (Africa)6% (Americas) to 5417% (Europe and C. Asia & Asia-Pacific) and this broadly reflects the different data publishing policies in these regions. As can be seen in Fig. 12, we recognise that the global distribution of data is sub-optimal, and we investigate the imbalance further in the discussion (Sect. 4.2). As discussed in Sect. 2.4.1, the data from Edmonds et al. (2020) consists of polygons drawn from four maximal extent points to create a four-sided polygon which~~

Of the leveed area data in openDELvE, 90% of the leveed area dataset by area (1,641 unique features) is considered of excellent or good quality (Fig. 3). This indicates that most of our sources are vector and high quality raster data, which we believe supports high quality onward data propagation to, and consumption by, the hydrological and risk modelling communities.

| Continental Zone (UN Region) | Number of deltas with levee data present in openDELvE[a] | Total number of unique leveed areas represented in dataset | Total geomorphic minimum deltaic area[a] [km²] | Area protected by levees within the delta[a] [km²] | Coverage of delta area by levees as % of deltaic area (computed) |
|---|---|---|---|---|---|
| Africa | 3 | 9 | 4,3524,3584,766 | 56957069 | 1323 % |
| Americas | 100 | 301 | 105,76699,262258,991 | 13,00015,282344 | 1215 6 %% |
| Asia-Pacific | 19 | 83 | 130,000128,970148,121 | 2,11025,396402 | 220 17 % |
| Europe & C. Asia | 310 | 6886 | 6,4926,45414,755 | 2,5603,4862,560 | 54 1397 % |
| **Processed Total** | **1532** | **461479** | **246239,044426,633885** | **44,73441,3993,875** | **19 107 %** |
| No Result | 1,0978 | - | 519,039502,928124,736 | - | - |
| Unprocessed (Pending & Not Processed) | 924 | - | 105,96413,402108,218 | - | - |
| **Global Total** | **2,174** | **47961[a]** | **874,142847,936564,771[b]** | **41,39944,73443,875** | **5 58 %** |

**Table 5: Summary of processed features and deltaic area at openDELvE release v1.0(current release) per geographic region and area totalled, figures are rounded to nearest whole integer.**

[a]**openDELvE contains 1,601 leveed area polygons but they are partially overlapping mainly due to the structure of administrative units in the USACE NLD. Overlapping sections are therefore are only counted once and for the purpose of this table article we 'dissolved' the layer to ensure that area was not double counted.**

[b]**Calculated by dissolving leveed area dataset to a single layer and compared to supplementary dataminimum delta geometry from Edmonds et al. (2020).**

Figure 32: Interactive browsing interface to openDELvE built using the ArcGIS Online platform. Area of focus is the Ganges-Brahmaputra delta, Bangladesh. Data content as per openDELvE version 1.0. Available publicly at: http://opendelve.eu

[Figure]

Figure 4<s>3</s>: **(a) Distribution of data quality classification in openDELvE <s>(v1.0)</s> given for each individual leveed area feature, classified according to the data quality matrix (Table 3). (b)**

<s>Figure 5</s>4: **Distribution of leveed area data in openDELvE <s>(v1.0)</s> by UN Region.<s> using allocation of deltas to region by Edmonds et al. (2020)</s>**

3.2 <s>openDELvE potential</s> Demonstrative applications of openDELvE

<s>The inclusion of</s>of the dData on levees <s>in</s>levee data can bring important insights and more accurate predictions in delta studies. Levees are sometimes included in small-scale studies, but not yet in large-scale or <s>(</s>global<s>)</s> studies (e.g. Dullaart et al., 2021, Nienhuis and van de Wal., 2021). Global studies are becoming more common, in part because of global challenges such as climate change (IPCC, 2021).

Here we showcase uses of openDELvE, including flood-protection of land use (what type of land is protected and what will be at risk), flood-protection of delta population (how many people live in flood-protected vs flood-prone areas), and potential improvements of flood hazard models in deltas (Figure 5).

<s>To sed</s>

[Figure]

**Figure <s>6</s>5: Examples of land use (Copernicus Global Land Service, Buchhorn et al., 2020), population (LandScan, https://landscan.ornl.gov/), and flooded area (Dullaart et al., 2021) within and outside levees <s>Side-by-side comparison of MODIS and LandScan™ data showing land use and urban zones</s> in the Mississippi Delta**

[Figure]

**Figure 6: Average land cover for 153 deltas within flood-protection levees compared to deltas as a whole.**

First, an intersection between openDELvE and land use data shows that  land-use patterns  differ significantly inside leveed areas compared to the rest of the delta ( Figure 5, Figure 7̶6). Urban and built-up land are concentrated within leveed areas, whereas wetlands and water bodies are more likely to be found outside levees. For example, 48% of flood-protected delta area is used as cropland, compared to 10% of the non-flood-protected  delta area  (Figure 6). Over- and under representation of different land use classes is likely because  L̲levees̶d are constructed to protect land with higher value̶d , such as  urban,  built-up areas and croplands. Levees are therefore important for food availability and access (Islam and Al Mamun, 2020), protection of urban centres and urban infrastructure (Jongman et al., 2012), and for reducing exposure to flooding (Lumbroso et al., 2017). There is a second effect that can also play a role. The existence of levees could lead to greater investment and development of urban and agricultural land compared to areas outside levees (Hutton et al., 2019), the so-called "levee effect". openDELvE does not include a̶ year of construction for levees, so that it is not possible to separate these two effects. ~~Meanwhile, water bodies and permanent wetlands which are not typically protected by levees.openDELvE was also combined with other existing datasets to indicate how it can be used in hazard modelling. Our analysis of population data within our subset of deltas (the openDELvE processed polygons) in section 3 supports the concept of the levee effect, where buildings and heavily urbanised zones are concentrated within leveed areas (see Figure 6) and this can result in the proliferation of buildings in such zones (see Figure 7).~~

~~We do however acknowledge thatWhilst our dataset is limited by the existence of data, however despite the limited number of deltas currently available, the concept of higher urbanisation being reflected in the leveed areas remains valid. Historically, in agricultural-rich zones such as the Mississippi Delta (Figure X)6) levee breaches have resulted in mass outward migration and an unexpected link into modernisation of historic process, as in the case of the 1927 Mississippi flood. (Hornbeck and Naidu, 2014)However with increasing percentages of urban settlement, this leads to a changed outlook on the future management of~~

~~Figure 7: Percentage difference between the modal land use for leveed areas versus total delta area for deltas processed in openDELvE (~ 150 deltas). Colour of the bar matches the MODIS scheme (see Figure 6).~~

[Figure]

**Figure 7: Flood-protected delta area vs flood-protected delta population, both as a fraction of the delta total. GB is the Ganges-Brahmaputra. The dotted line indicates the global average (GA).**

Second, our analysis with openDELvE and population data suggests that, for the 153 deltas in openDELvE, 74% of delta population lives outside flood-protected areas. Population densities are higher inside flood-protected areas: population dynamics and patterns also differ inside leveed delta regions compared to the whole delta averages. leveed areas occupy 17% of delta area, on average, but protect 26% of delta inhabitants. However, these global averages hide large differences between deltas and regions (Figure 7). In Europe (85%) and the Americas (41%) we find  a s relatively small leveed  e.g., the  Rhine-Meuse  delta in the Netherlands (92%), and the Mississippi in the USA (57%, Figure 5). This is not the case across Africa (3%) and Asia-Pacific (24%). Looking at population densities the pattern is different. In Asian  deltas, 800 people per km$^2$ live outside flood-protected areas, compared to 15,000 people per km$^2$ inside. In contrast, i Europe and the Americas there is only a 5 and 9 fold increase in population densities within flood-protected areas, respectively. The different patterns could be the result of competing factors in the co-evolution of levees and cities. the relationship between leveed area and the amount of people protected by levees mostly follows a 1:1 relation, meaning the population density within levees is approximately equal to the population density outside levees.

Although levees are constructed to protect people and are therefore expected to protect the most populated areas, they are also constructed in vulnerable delta locations, away from likely locations of major cities in regions that historically did not build levees (e.g., Bangladesh).

[Figure]

**Figure 8: Delta area potentially protected against coastal floods (in red), compared to all exposed delta area (in blue), for all deltas (left) and individual deltas (right). GB in the Ganges-Brahmaputra.**

~~Figure 8: Scatter plot indicating the relationship between percentage of the total delta population living withing leveed area and percentage leveed area per delta. Each delta processed in openDELvE (~ 150 deltas) is represented with colour indicating the UN region and the dashed line is the 1:1 relation.~~

In a third demonstration of openDELvE use, we assess the intersection of levees with global coastal flood assessments (Figure 5, Dullaart et al., 2021). When neglecting the presence of levees it would seem that 13% (32,261 km$^2$) of the combined area of the 153 deltas is exposed to coastal floods with return period of 10 years (Figure 8). This increases to 26% (63,179 km$^2$) and 39% (95,879 km$^2$) for 100-year and 1000-year floods, respectively. However, when accounting for  levees in  openDELvE, we find that these could reduce flood exposure by 25% 8,206 km$^2$ in the case of 10-year floods, and by 24% 22,744 km$^2$ in the case of 1000-yr floods (Figure 8). Protection against floods varies greatly between deltas. For the Rhine delta it is 78%, in the  Ganges-Brahmaputra delta it is 29% the Mississippi delta it is 13%  (Figure 5, Figure 8). Since openDELvE does not include data on levee heights and levee protection standards, we cannot associate each levee with a magnitude of flood; therefore, these numbers represent an approximation of the best-case protection offered by levees

~~Moreover, our analysis indicates that, if we assume that levees hold, the amount ea maps of people at risk in deltas can be significantly lower than hazard modelling without levees indicates. In deltas where levees were present, the flooded area can be decreased on average by 20-65%. In particular, levees significantly decrease predicted flooded area in 1:100 floods. The predicted flooded area is more accurate however, for 1:1000 year floods.~~

Figure 9: Box and whisker plot indicating the decrease in flooded area due to the presence of levees for 10,100 and 1000 year return period floods for deltas with levees processed in openDELvE

4. 4.0 Discussion

**4.1 How representative is openDELvE?**

As summarised in Table 5, we found that 197% of the geomorphic the delta area (which can include the shallow marine portions of the delta front, Edmonds et al., 2020) processed in openDELvE is protected by a levees. This should be considered
a rough estimate. For deltas covered by nationally maintained databases (e.g., Mississippi, Rhine-Meuse) the data quality is good. There, tThere is rich metadata and there is little chance of false negatives (no levee in openDELvE but missing existing levees present nevertheless). Data quality and coverage in other deltas (e.g. Ganges-Brahmaputra, Mekong) is poorer, and this appears to be linked to the lack of a nationally or regionally coordinated platform for levee data sharing. While individual levees or leveed areas are represented, tThere, the is a high chance is higher of false negatives and therefore undercounting
for the delta as a wholeof leveed area.

[revised manuscript text omitted]

---

## Author Response (AR2)

Reply to the editor and reviewers (in *italics*), line numbers refer to the tracked-changes manuscript text below.

**Editor Lindsay Beevers**

Dear authors

5 Please see the review comments from the two external reviewers. They are both in favour of acceptance after minor revisions. These revisions will then be reviewed by the editor before publication. Please pay attention to the minor points they raise and edit the manuscript in line with these. I very much look forward to receiving the revised manuscript.

Kind regards

10 Lindsay

> *Dear Lindsay,*
>
> *Thank you for your positive comments and the opportunity to provide a revised manuscript. We have addressed all review comments, and provided responses below.*
>
> *Thank you for your consideration,*
>
> 15 *Jaap Nienhuis, also on behalf of co-authors Jana Cox, Joey O'Dell, Douglas Edmonds, and Paolo Scussolini*

**Reviewer #3**

The authors present an interesting database on flood protection structures regarding the use of levees at a global scale. As an exercise, taking this macroscopic perspective is very useful and quite ambitious - 20 going through the thought process within the article is valuable. I appreciate that the authors have already made a lot of improvements already - However, I outline below some of my comments on what is a promising study.

> *We thank the reviewer for their review, and for their positive words about our paper.*

- Some figures could be more informative to make the study more engaging; In particular, Fig 2 and 3 - 25 the authors miss the opportunity to provide useful insights about the platform, and the information on levees that the reader should focus on; Some annotations there indicating on the levee location/extents and how some areas where defined would be very useful - at the moment these are not as informative as one would expect - i.e. I am not sure on what to focus on other than the platform shows some geospatial data.

*We appreciate these helpful comments. We have edited figures 2 and 3, and added further annotations. The new and old figures can be viewed side-by-side in the tracked-changes version of the manuscript, pasted below.*

*In figure 2 (line 255), we added details about the number of deltas within each case. We added the delta names, and explanation about the polygons can be found on the side panels.*

*In figure 3 (line 260) as well, we annotated the major items in the display: the Ganges-Brahmaputra Delta area, with leveed areas defined within the delta boundaries, and outside the boundaries. We further show a delta where no levees could be found, and have annotated the delta index on the top right.*

*We hope these changes have made the figures more informative and engaging.*

- One of the important findings that the authors stress regards the 17% of habitable area being confined within flood-protected levees as a key finding to focus on. The impression is that this is the result of overlaying the levee-protected geospatial areas against predictions from the global ocean model results. The issue here is that the global ocean model dataset has an 30 arcsec resolution (1km), and includes some pretty substantial assumptions on projecting coastal water levels in deltas and further inland - also, some more information about the ocean model accuracy and specifications would be useful. This is based on the fact that most global ocean models have some severe limitations in capturing coastal water levels, key hydrodynamics are not resolved, particularly when it comes to estuarine regions - and I suspect this adds substantial uncertainties in these findings. At the very least a discussion on this is required, and perhaps cross-validation with water elevation gauge data where in well monitored deltas would add some confidence in the analysis.

*We think there may be a misunderstanding here. The delta habitable area is not an output of a global ocean model and therefore also not necessarily limited by any resolution problems. The delta habitable area is manually retrieved by Caldwell et al (2019). It does includes some pretty substantial assumptions, which we now further clarify, but these are unrelated to coastal ocean water levels. Cross-validation with water elevation gauge data is a good idea, and has been done by the original authors (Dullaart et al) from whom we use the coastal flooding data.*

*We have added a section in the discussion (4.3, line 403) on the uncertainty in our levee data, the delta data, and the additional flooding, population, and land use data. We have also expanded the section in the discussion where we explain the 17% (section 4.1, line 370). Both can be found in our tracked-changed manuscript and also in our new polished manuscript.*

Overall the authors have made a good effort to present an interesting database - Some more thought on the uncertainty of their analysis through section 2.8 would add value - it seems that the authors seek to

demonstrate the substantial value of having such a database well-maintained, however the analysis there is a bit light touch and a more in-depth interpretation of the associated uncertainty in these statistics would strengthen the impact of this submission.

65

*We thank the reviewer for their review, and we have added the requested items.*

[revised manuscript text omitted]